# Knowledge, attitude and practice of wearing mask in the population presenting to tertiary hospitals in a developing country

Shumaila Furnaz[1]*, Natasha Baig[2], Sajjad Ali[3], Sahar Rizwan[4], Uzzam Ahmed Khawaja[5], Muhammad Abdullah Usman[3], Muhammad Tanzeel Ul Haque[3], Ayesha Rizwan[4], Farheen Ali[1], Musa Karim[1]

1 National Institute of Cardiovascular Diseases, Karachi, Pakistan, 2 Ziauddin Medical University, Karachi, Pakistan, 3 Dow Medical College, Karachi, Pakistan, 4 Jinnah Sindh Medical University, Karachi, Pakistan, 5 Jinnah Medical and Dental College, Karachi, Pakistan

* shumailafurnaz41@gmail.com

**Data Availability Statement:** All relevant data are within the paper and its Supporting information files.

## Abstract

### Background

In the era of COVID-19 where there is emphasis on the importance of wearing a mask, wearing it rightly is equally important. Therefore, the purpose of this study was to assess the knowledge, attitude and practice of wearing a mask in the general population of a developing country at three major tertiary care hospital.

### Materials and methods

Participants of this cross-sectional study were patients and attendants at three major tertiary care hospital of Karachi Pakistan. Selected participants, through non-probability convenient sampling technique, were interviewed regarding knowledge, attitude, and practice of wearing mask using an Urdu translated version of a questionnaire used in an earlier study. Three summary scores (0 to 100) were computed to indicate participants' mask wearing practice, technique of putting it on, and technique of taking if off. Collected data were analyzed with the help of IBM SPSS version 19.

### Results

A total of 370 selected individuals were interviewed, out of which 51.9% were male and mean age was 37.65±11.94 years. For more than 90% of the participants, wearing a face mask was a routine practicing during the pandemic. The mean practice score was 65.69±25.51, score for technique of putting on a face mask was 67.77±23.03, and score of technique of taking off a face mask was 51.01±29.23. Education level of participant tends to have positive relationship with all three scores, while presence of asthma or chronic obstructive pulmonary disease (COPD) as co-morbid had negative impact on mask wearing practice.

### Conclusion

We have observed suboptimal knowledge, attitude and practice of wearing mask among the selected individuals. There is a continued need to spread awareness and educate general

**Funding:** The authors received no specific funding for this work.

**Competing interests:** The authors have declared that no competing interests exist.

population about the importance of using a face mask, as well as the proper technique of wearing and taking off a face mask.

## Introduction

Face masks are a one-time use, affordable means to form a mechanical barrier from irritants and contagious diseases like airborne infections, hence, protecting from respiratory illnesses. They have demonstrated a pivotal role in curtailing the spread of the current Coronavirus Disease 2019 (COVID-19) pandemic. COVID-19, a respiratory disease caused by increased acute respiratory coronavirus syndrome 2 (SARS-CoV-2), initially appeared in China in December 2019 and has since spread to most countries worldwide, culminating in the COVID-19 pandemic [1]. As of today COVID-19 facts about Pakistan are as following; according to official figures total confirmed reported cases are 1,291,467, deaths are 28,878, and 60,043,930 individuals are fully vaccinated with the recommended dose of available COVID-19 vaccine [2].

The virus spreads primarily through respiratory droplets formed when an infected person coughs or sneezes, or by contacting contaminated surfaces or items, and then touching his or her mouth, nose, or possibly eyes. In their pandemic strategies, several nations, including Pakistan, have employed the use of face masks. The face mask works by creating a physical barrier in the immediate environment between the wearer's mouth and nose and possible pollutants [3]. In terms of the viral infectious dose, the reduction in exposure by wearing a mask could reduce the risk of infection [4] and, as a result, the transmission of the virus in the general population. Masks can play at least two roles in the prevention of viral transmission in the general population. Firstly, masks affect the formation of turbulent gas clouds and the emission of respiratory pathogens [5]. Research shows that masks can either block fast turbulent jets produced by coughing or redirect jets in much less harmful ways to manage airborne infections [6]. Secondly, the mask material can filter viral particles, like aerosols or droplets [7]. Appropriate donning and doffing steps should be followed to make the face mask more effective, as improper usage could increase the toll of these infections [8].

The proportion of people wearing masks during a pandemic depends on various factors, with society playing a crucial role in deciding the coverage of masks around the world [9]. In East Asia, wearing a mask is popular and has been culturally appropriate for a long time [9]. People wear masks for several purposes such as pollution, allergies, and winter safety, not just when they are sick. According to a recent Mintel survey, 63% of Japanese citizens wear face masks in public during the spread of COVID-19 [10]. However, public health authorities in North America and Europe have prevented safe people from wearing masks [9]. Previous studies across five countries highlighted a significant gap between willingness (71%) and real action (8%) to wear a mask in the United States [11]. One particular study advised imposing mandates to reinforce mask-wearing amongst the general population [12]. It could be an effective measure to control the surge in COVID-19 cases [12]. Furthermore, a study conducted in Civil Hospital, Karachi among healthcare workers showed limited knowledge, attitude, and practice of wearing a face mask [13].

The best way to wear a surgical mask is to wear the colored side facing out, independently of your health status, according to Nawhen, a columnist for Medical Myth busters Malaysia, a non-governmental organization that aims to address misconceptions and misleading evidence on medical matters. The hydrophobic outer layer is fluid-repelling, and its primary purpose is to keep germs from adhering to it. In contrast, the inner layer is a hydrophilic layer that

absorbs moisture from the air that is exhaled. The moisture from the air will bind to it if worn the other way round, thereby making it possible for germs to linger there. A middle layer directly filters the microorganism [14]. Thus, proper knowledge of how to wear a mask and its importance is imperative among the general population. A very limited literature are available on comprehension level of the information on importance and use of face mask among general population of developing and under-developing countries. In the era of COVID-19 where there is emphasis on the importance of wearing a mask, wearing it rightly is equally important. Therefore, the purpose of this study was to assess the knowledge, attitude and practice of wearing a mask in the general population of a developing country at three major tertiary care hospital.

## Material and methods

### Study design and setting

This cross-sectional study was conducted after the approval of the ethical review committee of National Institute of Cardiovascular Disease (NICVD), Karachi, Pakistan. This multicenter study was conducted at three major tertiary care hospital of Karachi Pakistan, namely Jinnah Postgraduate Medical Centre (JPMC), NICVD, and Dow University of Health Sciences (DUHS), between March 2021 and May 2021.

### Study population

Participants of this cross-sectional study were patients and attendants at three major tertiary care hospital of Karachi Pakistan. Intercept study participants were recruited through non-probability convenient sampling technique from the waiting area of the outpatient clinics. Verbal informed consent was obtained from all the participants. All the individuals fulfilling the inclusion criteria were taken to a dedicated booth for the interview. Inclusion criteria for the study were either gender, aged $\geq$ 18 years, and presented in hospital premises as patient or attendants, hospital employees were not included in this study. Face to face interviews were conducted by the trained healthcare workers and both interviewer and respondents were strictly adherent to the COVID-19 standard operating procedures (SOPs) which included hand sanitization, wearing a face mask, and keeping a safe distance.

### Study instrument

An interview regarding knowledge, attitude and practice of wearing a mask was conducted with all the participants using the Urdu translated and regional adopted questionnaire used by Lee LYu et al. [15]. The Urdu translated version of the questionnaire was internally validated for the validity and relevance of the content to the context of mask wearing habit in local scenario of Pakistani population. The questionnaire consisted of three domains, first domain was regarding assessment of participants' practice of wearing a face mask in given scenarios where wearing a face mask was declared necessary. Each scenario was defined in a statement and participant's adherence to the mask wearing practice in given scenario was rated on three-point frequency scale as "never", "sometimes", and "always". 'Not applicable' indicated that the participants did not encounter a situation that required wearing a face mask. An aggregated percentage score was computed by assigning percentage point to each of the response as "0" to never, "50" to sometimes, and "100" to always rating. Response "not applicable" was excluded from the calculation of aggregated percentage score.

Second domain was regarding assessment of face mask wearing technique in which 12 face mask wearing steps were outlined and response of the participant was recorded as "yes" and

"no". The final domain was regarding assessment of face mask taking off technique in which 8 steps were outlined and participant response was recorded as "yes" and "no". Aggregated mask wearing technique and mask taking off technique score were computed by assigning 100 points to each "yes" response and 0 points to each "no" response. Dataset along with the detailed questionnaire (in English and Urdu) are provided as S1 Data and S1 Questionnaire respectively.

## Data analysis

In the absence of relevant literature on mask wearing habit it our population, the sample size for the study was calculated with anticipated proportion of mask wearing practice as 50%, at 95% confidence level, and 5% error margin the required sample size of 384 was calculated. Statistical Package for the Social Sciences (IBM SPSS) Version 21.0 was used for data analysis. No missing value imputation methods were used and analysis were performed on complete dataset of 370 responses after excluding 14 responses due to missing values. The common method bias in the responses to the attitude and practice related attributes was assessed by conducting Harman's single factor test and Kaiser–Meyer–Olkin (KMO) test. Collected data were summarized with the help of descriptive statistics such as mean ± standard deviation (SD) or percentage (frequency). Face mask wearing practice and non-practicing group of participants were compared for various demographic and clinical characteristics and mask wearing practice and techniques attributed by applying Chi-square test. Differences in the mean score of three domains by various demographic characteristics were assessed with the help of independent sample t-test. The significance level was $\leq 0.05$.

## Results

A total of 370 participants successfully completed the interview questionnaire. 51.9% were male and the mean age of the participants was 37.65 ± 11.94 years, the majority (74.9%) of whom were under 45 years of age. More than 90% of the participants were practicing face mask wearing in their day to day life during the COVID-19 pandemic. A mask wearing habit positively associated with the participants' education level and negatively associated with the presence of asthma or Chronic Obstructive Pulmonary Disease (COPD) as co-morbid conditions. Demographic characteristics and risk profile stratified by the face mask wearing habit of the respondent are presented in (Table 1).

The KMO measure of sampling adequacy was 0.676 with Harman's single factor test for common method bias showing 20.96% variance explanation ensuring both adequacy and bias free information for the analysis. Participants were found to adhere more to face mask wearing practices while visiting hospitals (88.1% respondent reported frequency of 'Always') and clinics during the pandemic (69.5% respondent reported frequency of 'Always'). The practice of wearing a mask was lesser reported for situations like taking care of family members with fever (34.3% respondent reported frequency of 'Never') and taking care of family members with a respiratory infection (28.1% respondent reported frequency of 'Never'). The distribution of the practice of using a face mask specific to various situations is reported in (Fig 1).

The stepwise techniques of putting on and taking off a face mask followed by the respondents are reported in Fig 1. Hand hygiene before wearing (31.1%) and before taking off a face mask (21.9%) were the least followed steps. Less than 50% of the participants reported performing hand hygiene after disposing off the face mask.

Mean percentage score of practice of mask wearing in various situations and technique of putting on and taking off face mask by various demographic and baseline characteristics are presented in (Table 2).

**Table 1. Demographic characteristics and risk profile stratified by the face mask wearing habit of the respondent.**

| Characteristics | Total | Mask wearing habit during COVID-19 pandemic | | P-value |
|---|---|---|---|---|
| | | No | Yes | |
| **Total (N)** | **370** | **30 (8.1%)** | **340 (91.9%)** | - |
| **Gender** | | | | |
| Male | 51.9% (192) | 8.3% (16) | 91.7% (176) | 0.869 |
| Female | 48.1% (178) | 7.9% (14) | 92.1% (164) | |
| **Age (years)** | 37.65 ± 11.94 | 39.6 ± 15.94 | 37.48 ± 11.54 | 0.482 |
| 18 to 45 years | 74.9% (277) | 7.6% (21) | 92.4% (256) | 0.522 |
| > 45 years | 25.1% (93) | 9.7% (9) | 90.3% (84) | |
| **Residence** | | | | |
| Urban | 65.4% (242) | 6.6% (16) | 93.4% (226) | 0.147 |
| Rural | 34.6% (128) | 10.9% (14) | 89.1% (114) | |
| **Socioeconomic class (PKR/month)** | | | | |
| Low (≤ 25000) | 54.3% (201) | 9.5% (19) | 90.5% (182) | 0.440 |
| Middle (25000–50000) | 40.8% (151) | 6% (9) | 94% (142) | |
| High (≥ 50000) | 4.9% (18) | 11.1% (2) | 88.9% (16) | |
| **Education** | | | | |
| Low (primary or less) | 37.6% (139) | 13.7% (19) | 86.3% (120) | **0.010**[*] |
| Middle (up to 10th grade) | 49.2% (182) | 4.9% (9) | 95.1% (173) | |
| High (12th grade or higher) | 13.2% (49) | 4.1% (2) | 95.9% (47) | |
| **Hypertension** | | | | |
| No | 58.1% (215) | 9.8% (21) | 90.2% (194) | 0.168 |
| Yes | 41.9% (155) | 5.8% (9) | 94.2% (146) | |
| **Diabetes** | | | | |
| No | 61.6% (228) | 9.6% (22) | 90.4% (206) | 0.169 |
| Yes | 38.4% (142) | 5.6% (8) | 94.4% (134) | |
| **Current Smoker** | | | | |
| No | 74.6% (276) | 7.2% (20) | 92.8% (256) | 0.298 |
| Yes | 25.4% (94) | 10.6% (10) | 89.4% (84) | |
| **Asthma/COPD** | | | | |
| No | 89.2% (330) | 7% (23) | 93% (307) | **0.021**[*] |
| Yes | 10.8% (40) | 17.5% (7) | 82.5% (33) | |
| **Ischemic heart disease** | | | | |
| No | 84.1% (311) | 9% (28) | 91% (283) | 0.148 |
| Yes | 15.9% (59) | 3.4% (2) | 96.6% (57) | |
| **Family COVID-19 history** | | | | |
| No | 81.9% (303) | 7.9% (24) | 92.1% (279) | 0.779 |
| Yes | 18.1% (67) | 9% (6) | 91% (61) | |
| **Exposed to COVID-19 patients** | | | | |
| No | 79.5% (294) | 6.8% (20) | 93.2% (274) | 0.070 |
| Yes | 20.5% (76) | 13.2% (10) | 86.8% (66) | |

[*]significant at 5%.

COPD = chronic obstructive pulmonary disease, COVID-19 = coronavirus disease.

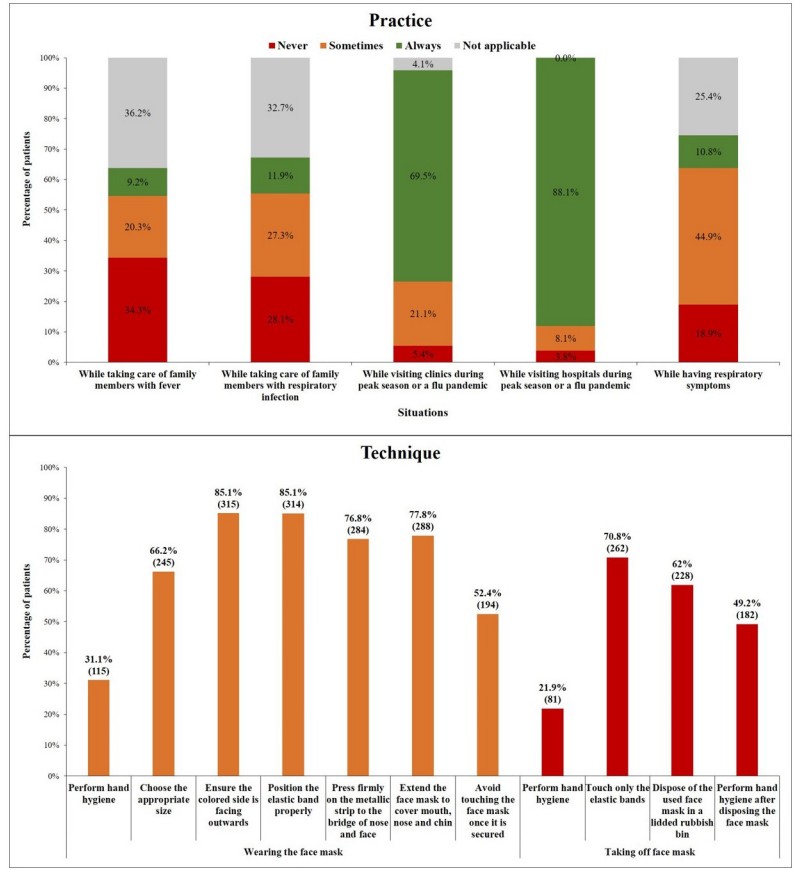

**Fig 1. Distribution of practice of using face mask specific to various situations and technique of putting on and taking of face mask.**

Mean percentage score of practice of mask wearing in various situations was 65.69 ± 25.51, mean percentage score for technique of putting on face mask was 67.77 ± 23.03, and mean percentage score for technique of taking of face mask was 51.01 ± 29.23. Education level of the respondent tends to have positive relationship with all three scores, while presence of asthma or COPD as co-morbid had negative impact on mask wearing practice.

## Discussion

The general population presenting to tertiary care hospitals in Karachi, Pakistan were not using face masks optimally. The importance of hand hygiene as an essential step in wearing and taking off a mask was deeply undervalued. Considered the importance of subject matter, both practice and technique related to face mask use were suboptimal in our population. Education level of participant tends to have positive relationship towards face mask practice and technique while presence of some co-morbid conditions such as asthma or COPD had negative impact on mask wearing practice. Although, no prior literature regarding mask wearing practice are available for our population, but the lack of adherence to the protocols of mask wearing in our populations was same as the observation made by Lee LY et al. [15] among adult population of Hong Kong. A separate discussion on each of the findings is presented in the following sections.

**Table 2. Mean percentage score of practice of mask wearing and technique of putting on and taking off face mask by various demographic and baseline characteristics.**

| Demographic characteristics | Practice of mask wearing | Technique of putting on face mask | Technique of taking off face mask |
|---|---|---|---|
| **Gender** | | | |
| Male | 69.42 ± 25.69 | 66.25 ± 22.9 | 50.52 ± 28.48 |
| Female | 61.68 ± 24.76 | 69.42 ± 23.12 | 51.54 ± 30.08 |
| *P-value* | ***0.003**** | *0.185* | *0.737* |
| **Age (years)** | | | |
| 18 to 45 years | 65.26 ± 25.75 | 67.94 ± 23.13 | 51.17 ± 29.66 |
| > 45 years | 66.97 ± 24.87 | 67.28 ± 22.84 | 50.54 ± 28.07 |
| *P-value* | *0.577* | *0.812* | *0.856* |
| **Residence** | | | |
| Urban | 66.31 ± 25.99 | 67.49 ± 23.73 | 52.17 ± 28.86 |
| Rural | 64.52 ± 24.62 | 68.3 ± 21.73 | 48.83 ± 29.9 |
| *P-value* | *0.522* | *0.748* | *0.296* |
| **Socioeconomic class (PKR/month)** | | | |
| Low (≤ 25000) | 63.73 ± 26.86 | 67.45 ± 22.96 | 51.24 ± 30.9 |
| Middle (25000–50000) | 68.32 ± 23.38 | 68.34 ± 24.02 | 50 ± 26.77 |
| High (≥ 50000) | 65.65 ± 26.55 | 66.67 ± 14.7 | 56.94 ± 30.69 |
| *P-value* | *0.248* | *0.918* | *0.628* |
| **Education** | | | |
| Low (primary or less) | 60.64 ± 26.49 | 65.16 ± 24.99 | 47.12 ± 30.26 |
| Middle (up to 10th grade) | 68.21 ± 23.45 | 68.08 ± 22.5 | 51.1 ± 29.12 |
| High (12th grade or higher) | 70.68 ± 28.08 | 74.05 ± 17.66 | 61.73 ± 23.99 |
| *P-value* | ***0.010**** | *0.065* | ***0.010**** |
| **Hypertension** | | | |
| No | 69.88 ± 26.77 | 68.06 ± 21.64 | 52.67 ± 29.63 |
| Yes | 59.89 ± 22.47 | 67.37 ± 24.89 | 48.71 ± 28.6 |
| *P-value* | ***0.001**** | *0.777* | *0.198* |
| **Diabetes** | | | |
| No | 67.64 ± 27.4 | 67.13 ± 23.73 | 51.1 ± 30.31 |
| Yes | 62.58 ± 21.86 | 68.81 ± 21.9 | 50.88 ± 27.52 |
| *P-value* | *0.063* | *0.494* | *0.945* |
| **Current Smoker** | | | |
| No | 65.74 ± 25.81 | 69.62 ± 22.77 | 52.45 ± 29.48 |
| Yes | 65.56 ± 24.75 | 62.36 ± 23.06 | 46.81 ± 28.22 |
| *P-value* | *0.953* | ***0.008**** | *0.106* |
| **Asthma/COPD** | | | |
| No | 66.87 ± 25.75 | 68.85 ± 22.38 | 51.82 ± 28.79 |
| Yes | 56 ± 21.34 | 58.93 ± 26.52 | 44.38 ± 32.27 |
| *P-value* | ***0.011**** | ***0.010**** | *0.128* |
| **Ischemic heart disease** | | | |
| No | 67.25 ± 26.21 | 68.23 ± 22.78 | 50.4 ± 29.48 |
| Yes | 57.5 ± 19.67 | 65.38 ± 24.34 | 54.24 ± 27.91 |
| *P-value* | ***0.007**** | *0.384* | *0.356* |
| **Family COVID-19 history** | | | |
| No | 66.76 ± 26.53 | 69.17 ± 22.85 | 51.16 ± 29.39 |
| Yes | 60.86 ± 19.7 | 61.48 ± 22.97 | 50.37 ± 28.7 |
| *P-value* | *0.086* | ***0.013**** | *0.843* |

*(Continued)*

**Table 2.** (Continued)

| Demographic characteristics | Practice of mask wearing | Technique of putting on face mask | Technique of taking off face mask |
|---|---|---|---|
| **Exposed to COVID-19 patients** | | | |
| No | 66.9 ± 26.18 | 69.06 ± 22.66 | 52.13 ± 29.53 |
| Yes | 61.03 ± 22.29 | 62.78 ± 23.91 | 46.71 ± 27.79 |
| *P-value* | *0.074* | ***0.034****\** | *0.150* |

\*significant at 5%.

COPD = Chronic Obstructive Pulmonary Disease, COVID-19 = Coronavirus Disease 2019.

## Gender and mask wearing

The practice of wearing a mask and gender were found to be statistically significant, whereas the technique of putting on and taking off a mask were not significant. This outcome has been supported by multiple studies in the past which state that males are more knowledgeable about wearing a mask than female participants [16, 17]. In a survey conducted on the general public of Pakistan, males were found to gather socially more often than females; a possible explanation of why they are more likely to wear a mask [17]. However, other research revealed that female respondents were more likely to wear a mask in their day-to-day life when going outside [18] and that women were more likely to have good practices regarding masks and other preventive measures [19]. Amongst a cross-section of young adults in Pakistan, females were more prone to overestimate the fatality of coronavirus and demonstrated more knowledge-seeking behavior than their male counterparts, reasons that could explain a higher likelihood of females wearing a mask [20].

It is important to note that men are more susceptible to coronavirus infection than women due to physiologic differences such as sex hormones and a higher expression of coronavirus receptors in males, as well as lifestyle differences such as increased tobacco and alcohol consumption by men [21]. Thus, the emphasis on wearing a mask, especially for the male population, is crucial and strict implementation measures must be taken worldwide.

## Level of literacy, socioeconomic status and mask wearing

A statistically significant relationship was noted between education levels and the practice of wearing a mask (p = 0.01), as well as technique of taking off a mask (p = 0.01). This notion has been well demonstrated in many past studies [16–18]. It is obvious that the use of the internet, digital and print media to spread awareness about COVID-19 is better understood and thus, practiced by the literate population. Additionally, these informative channels are more affordable for and accessible to the educated masses. Limited access to all sources of information makes vulnerable populations such as rural families more susceptible to having poor knowledge and inadequate practices towards the COVID-19 pandemic [18].

Moreover, other precautionary measures such as hand hygiene are also more prevalent in participants with a higher level of education [22]. Those using masks or practicing appropriate hand hygiene more than four times a day have been shown to have greater knowledge about coronavirus [23]. Lower education levels have also been highlighted as a predictor of poor practices of coronavirus prevention measures [16].

No statistical significance was noted between socioeconomic status and the prevalence of wearing a mask. However, it has been previously established that participants with higher income levels use N95 and surgical masks, while respondents belonging to lower income groups are more likely to wear cloth masks [17].

## Co-morbid conditions and mask wearing

Several co-morbid conditions namely Hypertension, Diabetes, Asthma, COPD and Ischemic Heart Disease (IHD) were included. On analysis, a statistically significant relationship between hypertension and wearing a mask, asthma and COPD and wearing a mask as well as technique of putting on a mask, and IHD and wearing a mask was discovered.

Surprisingly, there was a negative association between wearing a mask and the presence of asthma or COPD. One possible explanation for this may be that wearing a mask causes respiratory difficulties in those with already compromised pulmonary function. It has been proven that, in those suffering from COPD, the heart rate, breathing frequency and end-tidal PCO2 were significantly higher after a 6-minute walk test while wearing an N95 mask versus without an N95 mask, whereas the $SPO_2$ levels were significantly lower [24]. Based on these findings, it is advised that those suffering from COPD who have Medical Research Council dyspnea scale (mMRC) scores $\geq$ 3 or Forced Expiratory Volume in 1 second (FEV1) < 30% predicted should use N95 masks with caution due to an increased risk of inducing respiratory failure [24]. However, when using a surgical face mask, subjects with COPD have not shown significant physiologic changes in gas exchange measurements after a 6-minute walk test [25].

Patients with controlled or mild asthma can safely wear a surgical mask for four continuous hours during their usual activities [26]. The Asthma and Allergy Foundation of America state that patients with mild and/or well-controlled asthma are likely to wear a face mask comfortably, while for those with severe disease and frequent asthma-related issues, wearing a mask may cause issues [27]. Therefore, for those who are unable to wear a mask due to breathing difficulties, it is advised that they practice other coronavirus preventive measures such as staying at home, avoiding contact with sick individuals, maintaining a 6 feet social distance in public and practicing hand hygiene [27].

Similarly, a negative association between hypertension and the practice of wearing a mask, as well as IHD and the practice of wearing a mask was also revealed. This is alarming as it is unequivocal that increased blood pressure and cardiac injury put individuals at a higher risk of COVID-19 complications and mortality [28]. Therefore, it is essential to spread awareness about the importance of wearing a mask and practicing other coronavirus precautionary measures in these high-risk patient populations.

## Exposure to COVID-19 patients, family history of COVID-19 and mask wearing

Participants who had been exposed to COVID-19 patients were less likely to wear a mask. Additionally, those with a positive family history of COVID-19 also wore a mask less frequently than those with a negative family history.

The incubation period of COVID-19 before the manifestation of symptoms is between 2 days to 15 days [29]. It is hypothesized that people are most infectious at this stage, when symptoms are absent or mild [30]. Individuals who have been in close contact with a COVID-19 patient are most likely to fall in this category of asymptomatic people and hence, should be extra cautious to wear a mask. In a study conducted in Beijing, China, face masks proved to be 79% effective in reducing the transmission of coronavirus when used by the patient and their family contacts [31].

Even though this the first study of its kind in our population, main limitation of the study was limited sample size and hospital setup. Hence generalizability of study results to the entire general population may be limited.

## Conclusion

We have observed suboptimal knowledge, attitude and practice of wearing mask among the selected individuals. There is a continued need to spread awareness and educate general population about the importance of using a face mask, as well as the proper technique of wearing and taking off a face mask. Those with co-morbid conditions who are at the highest risk of COVID-19 mortality should be especially vigilant about following preventive measures. Moreover, individuals with limited access to news outlets as well as those with low literacy rates living in remote areas should receive special training in their own languages through the government on the use of face masks, especially in developing countries such as Pakistan.

## Supporting information

**S1 Data. Dataset in SPSS format.**
(SAV)

**S1 Questionnaire. Questionnaire with Urdu translation in PDF format.**
(PDF)

## Acknowledgments

The authors wish to acknowledge the support of the staff members of the Clinical Research Department of the National Institute of Cardiovascular Diseases (NICVD) Karachi, Pakistan.

## Author Contributions

**Conceptualization:** Shumaila Furnaz, Sahar Rizwan, Farheen Ali, Musa Karim.

**Data curation:** Shumaila Furnaz, Natasha Baig, Sajjad Ali, Sahar Rizwan, Uzzam Ahmed Khawaja, Muhammad Abdullah Usman, Muhammad Tanzeel Ul Haque, Ayesha Rizwan, Farheen Ali.

**Formal analysis:** Musa Karim.

**Investigation:** Sajjad Ali, Uzzam Ahmed Khawaja, Muhammad Abdullah Usman, Muhammad Tanzeel Ul Haque, Ayesha Rizwan, Farheen Ali.

**Methodology:** Shumaila Furnaz, Sahar Rizwan, Farheen Ali, Musa Karim.

**Project administration:** Shumaila Furnaz, Uzzam Ahmed Khawaja, Muhammad Tanzeel Ul Haque, Ayesha Rizwan.

**Resources:** Shumaila Furnaz, Sajjad Ali, Sahar Rizwan, Muhammad Abdullah Usman, Farheen Ali.

**Software:** Sajjad Ali, Ayesha Rizwan, Musa Karim.

**Supervision:** Shumaila Furnaz, Uzzam Ahmed Khawaja, Muhammad Abdullah Usman, Farheen Ali.

**Validation:** Natasha Baig, Sahar Rizwan, Muhammad Abdullah Usman, Ayesha Rizwan, Musa Karim.

**Visualization:** Natasha Baig, Uzzam Ahmed Khawaja, Muhammad Tanzeel Ul Haque, Farheen Ali.

**Writing – original draft:** Shumaila Furnaz.

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
