## [Decision Letter · Decision Letter 0]

13 Dec 2021

PONE-D-21-33953Knowledge, Attitude and Practice of Wearing Mask in the Population Presenting to Tertiary Hospitals in a Developing CountryPLOS ONE

Dear Dr. Furnaz,

Thank you for submitting your manuscript to PLOS ONE. After careful consideration, we feel that it has merit but does not fully meet PLOS ONE’s publication criteria as it currently stands. Therefore, we invite you to submit a revised version of the manuscript that addresses the points raised during the review process.

Kindly revise a manuscript and submit a revised version which will be reviewed again. 

We look forward to receiving your revised manuscript.

Kind regards,

Pathiyil Ravi Shankar

Academic Editor

PLOS ONE

Journal Requirements:

3. One of the noted authors is a group or consortium MSc, and MBBS. In addition to naming the author group, please list the individual authors and affiliations within this group in the acknowledgments section of your manuscript. Please also indicate clearly a lead author for this group along with a contact email address.

5. In your Supporting Information, you have not specified where the minimal data set underlying the results described in your manuscript can be found. PLOS defines a study's minimal data set as the underlying data used to reach the conclusions drawn in the manuscript and any additional data required to replicate the reported study findings in their entirety. All PLOS journals require that the minimal data set be made fully available. For more information about our data policy, please see http://journals.plos.org/plosone/s/data-availability.

"Upon re-submitting your revised manuscript, please upload your study’s minimal underlying data set as Supporting Information files or to a stable, public repository and include the relevant URLs, DOIs, or accession numbers within your revised cover letter. For a list of acceptable repositories, please see http://journals.plos.org/plosone/s/data-availability#loc-recommended-repositories. Any potentially identifying patient information must be fully anonymized.

We will update your Supporting Information to reflect the information you provide in your cover letter.

Reviewers' comments:

Reviewer's Responses to Questions

**Comments to the Author**

1. Is the manuscript technically sound, and do the data support the conclusions?

Reviewer #1: Yes

Reviewer #2: Partly

2. Has the statistical analysis been performed appropriately and rigorously? 

Reviewer #1: Yes

Reviewer #2: No

3. Have the authors made all data underlying the findings in their manuscript fully available?

Reviewer #1: Yes

Reviewer #2: No

4. Is the manuscript presented in an intelligible fashion and written in standard English?

Reviewer #1: Yes

Reviewer #2: No

5. Review Comments to the Author

Reviewer #1: It seems "authors have worked hard in conducting this study and writing the manuscript of study.

I would like to draw attention of authors on following:

1. The language of the purpose of the study must be same both in the abstract and in the end of introduction.

Abstract: "The purpose of this study was to assess the knowledge, attitude and practice of wearing a mask in the general population presented to three major tertiary care hospitals of a developing country."

Introduction: "The purpose of this study was to assess the knowledge, attitude and practice of wearing a mask in the general population of a developing country at two major public sector and one private sector hospital."

Methodology: Authors have mentioned "regional adopted questionnaire used"; they have to clarify with regard to the region.

Figures (1 and 2) are not properly labelled.

Conclusion: The sentence mentioned in the conclusion in abstract is not included in conclusion of text of manuscript. Authors have taken sample of the people (patients and others) visiting the hospitals but not the sample of the people in general population; the conclusion must be focused on people (patients and others) visiting the hospitals as hospitals usually don't allow the people enter into hospital without wearing mask. People visiting hospitals are bound to wear the mask but general public usually don't follow the instruction in Pakistan as observed.

Authors have also mention the limitation of the study.

Reviewer #2: Abstract: This needs to be written again, in its current form it is poorly written. In the background, the authors require to highlight the importance of wearing a mask before explaining the purpose. In material and methods authors needs to mention which sampling approach and analysis tool have been used. Results need correction such [technique of putting on was 67.77±23.03, and technique 36 of taking off was 51.01±29.23] shown incomplete sentence.

Introduction The flow of the introduction is generally poorly structured. They have not provided some evidence regarding COVID-19 cases, vaccines delivered in developing countries. In addition, there is a needs to report the research gap and why this study is needed. What are the contributions of the study?.

Methods: This section needs to be sub-divided into the sampling and data collection, recruitment of the respondents, measures of the study (what questions have been asked), and data analysis. Furthermore, there can be critiques on the items used whether those are validated or not. There are several other issues in this part of the study such as how data was collected (i.e., face to face, online), participants have been approached and sampled. Why did they only select 370 participants, how the sample was calculated, did they use any calculation formula, if the authors employed it, there is a need to be reported. Proper justification needs to be added.

Results: This section has several issues as well. Such as which analytical tools and techniques have been used. Why authors did not use hierarchal regression. In addition, there can be common method bias, there is a need to report Harman’s single factor, and KMO test to ensure data is free from any bias.

Discussion: In this section, there is a need to give more importance to insignificant results, general results have been discussed less attention has been given to prior studies. For instance, whether work is consistent or inconsistent with previous studies.

Conclusion: This section needs to be sub-sectioned into implications (general/practical and government/policymakers), limitations, and future directions of the study.

Apart from the above, there are several grammatical errors into the study such as refer line 36, taking of needs to be taking off.

6. PLOS authors have the option to publish the peer review history of their article (what does this mean?). If published, this will include your full peer review and any attached files.

Reviewer #1: **Yes: **Rano Mal Piryani

Reviewer #2: **Yes: **Sikandar Ali Qalati

---

## [Author Response · Author response to Decision Letter 0]

30 Dec 2021

Pathiyil Ravi Shankar

Academic Editor

PLOS ONE

30th December 2021

Dear Dr. Pathiyil Ravi Shankar,

We are respectfully submitting our revised manuscript titled “Knowledge, Attitude and Practice of Wearing Mask in the Population Presenting to Tertiary Hospitals in a Developing Country” [PONE-D-21-33953] - [EMID:9854e2ba05706459] for publication consideration in PLOS ONE. We appreciate the insightful editorial and reviewer comments, all of which have been specifically addressed in this revised version of the paper. Please find below the reviewer comments in bold and the author responses in non-bold. We believe these changes have strengthened the quality of our manuscript and that you will find it suitable for publication in the Journal. 

All authors had access to all the study data, take responsibility for the accuracy of the analysis, and had authority over manuscript preparation and the decision to submit the manuscript for publication. The manuscript represents original work that has not been published and is not under consideration for publication in any other journal. All authors meet the criteria for authorship and instructions to the author were read. We accept all conditions and publication rights. We have no conflicts of interest to declare and no funding sources to declare. 

Thank you for your consideration, and we hope that you find our revised manuscript suitable for publication in PLOS ONE.

Thank you,

Yours sincerely,

Shumaila Furnaz, Manager at Research Department, National Institute of Cardiovascular Diseases, Karachi, Pakistan.

E-mail: shumailafurnaz41@gmail.com

Mobile: +923452093086

Fax: +92-21-99201287

Journal Requirements:

Response: Formatting of the title page and body of the text as well as file naming are updated as per the PLOS ONE's style requirements

Response: There are no restrictions data availability, a minimal anonymized data set is uploaded in “.sav” format along with the revised manuscript

3. One of the noted authors is a group or consortium MSc, and MBBS. In addition to naming the author group, please list the individual authors and affiliations within this group in the acknowledgments section of your manuscript. Please also indicate clearly a lead author for this group along with a contact email address.

Response: There are no group authors and title page is updated accordingly

Response: captions for supporting information files are provided at the end of revised manuscript

5. In your Supporting Information, you have not specified where the minimal data set underlying the results described in your manuscript can be found. PLOS defines a study's minimal data set as the underlying data used to reach the conclusions drawn in the manuscript and any additional data required to replicate the reported study findings in their entirety. All PLOS journals require that the minimal data set be made fully available. For more information about our data policy, please see http://journals.plos.org/plosone/s/data-availability.

"Upon re-submitting your revised manuscript, please upload your study’s minimal underlying data set as Supporting Information files or to a stable, public repository and include the relevant URLs, DOIs, or accession numbers within your revised cover letter. For a list of acceptable repositories, please see http://journals.plos.org/plosone/s/data-availability#loc-recommended-repositories. Any potentially identifying patient information must be fully anonymized.

We will update your Supporting Information to reflect the information you provide in your cover letter.

Response: Reference to the supporting information is added to the methodology section at appropriate position

Reviewers' comments:

Reviewer's Responses to Questions

Comments to the Author

1. Is the manuscript technically sound, and do the data support the conclusions?

Reviewer #1: Yes

Reviewer #2: Partly

Response: Methodology section is updated to address the technical aspects of the study design 

2. Has the statistical analysis been performed appropriately and rigorously?

Reviewer #1: Yes

Reviewer #2: No

Response: Statistical aspects are elaborated in the methodology section.

3. Have the authors made all data underlying the findings in their manuscript fully available?

Reviewer #1: Yes

Reviewer #2: No

Response: Data availability statement is updated and a minimal anonymized data set is uploaded in “.sav” format along with the revised manuscript.

4. Is the manuscript presented in an intelligible fashion and written in standard English?

Reviewer #1: Yes

Reviewer #2: No

Response: Typographical or grammatical errors are corrected throughout the manuscript 

5. Review Comments to the Author

Reviewer #1: It seems "authors have worked hard in conducting this study and writing the manuscript of study.

I would like to draw attention of authors on following:

1. The language of the purpose of the study must be same both in the abstract and in the end of introduction.

Abstract: "The purpose of this study was to assess the knowledge, attitude and practice of wearing a mask in the general population presented to three major tertiary care hospitals of a developing country."

Introduction: "The purpose of this study was to assess the knowledge, attitude and practice of wearing a mask in the general population of a developing country at two major public sector and one private sector hospital."

Response: Study objective mentioned in the introduction section and abstract section are updated as per the suggestion

Methodology: Authors have mentioned "regional adopted questionnaire used"; they have to clarify with regard to the region.

Response: Details on regional adoption of questionnaire are added to the methodology section

Figures (1 and 2) are not properly labelled.

Response: Updated as per the suggestion

Conclusion: The sentence mentioned in the conclusion in abstract is not included in conclusion of text of manuscript. Authors have taken sample of the people (patients and others) visiting the hospitals but not the sample of the people in general population; the conclusion must be focused on people (patients and others) visiting the hospitals as hospitals usually don't allow the people enter into hospital without wearing mask. People visiting hospitals are bound to wear the mask but general public usually don't follow the instruction in Pakistan as observed.

Response: Updated as per the suggestion

Authors have also mention the limitation of the study.

Response: Study limitations are added as per the suggestion

Reviewer #2: 

Abstract: This needs to be written again, in its current form it is poorly written. In the background, the authors require to highlight the importance of wearing a mask before explaining the purpose. In material and methods authors needs to mention which sampling approach and analysis tool have been used. Results need correction such [technique of putting on was 67.77±23.03, and technique 36 of taking off was 51.01±29.23] shown incomplete sentence.

Response: Abstract is updated as per the suggestion

Introduction The flow of the introduction is generally poorly structured. They have not provided some evidence regarding COVID-19 cases, vaccines delivered in developing countries. In addition, there is a needs to report the research gap and why this study is needed. What are the contributions of the study?.

Response: As per the suggestions, statistics regarding COVID-19 cases and vaccines delivery in the local context are added to the introduction section with rationale of study highlighting the need of this study in our population.

Methods: This section needs to be sub-divided into the sampling and data collection, recruitment of the respondents, measures of the study (what questions have been asked), and data analysis. Furthermore, there can be critiques on the items used whether those are validated or not. There are several other issues in this part of the study such as how data was collected (i.e., face to face, online), participants have been approached and sampled. Why did they only select 370 participants, how the sample was calculated, did they use any calculation formula, if the authors employed it, there is a need to be reported. Proper justification needs to be added.

Response: As per the suggestions, methodology section is divided into sub-sections as study design and setting, study population, study instrument, and data analysis. Details on regional adoption of questionnaire are added to the methodology section. Details and assumptions used for the calculation of sample size are added. 

Results: This section has several issues as well. Such as which analytical tools and techniques have been used. Why authors did not use hierarchal regression. In addition, there can be common method bias, there is a need to report Harman’s single factor, and KMO test to ensure data is free from any bias.

Response: As per the suggestions, Statistical method section is updated regarding assessment of bias in the responses and results of Harman’s single factor test and Kaiser–Meyer–Olkin (KMO) test are provided in results section. 

Discussion: In this section, there is a need to give more importance to insignificant results, general results have been discussed less attention has been given to prior studies. For instance, whether work is consistent or inconsistent with previous studies.

Response: Updated as per the suggestion

Conclusion: This section needs to be sub-sectioned into implications (general/practical and government/policymakers), limitations, and future directions of the study.

Apart from the above, there are several grammatical errors into the study such as refer line 36, taking of needs to be taking off.

Response: Updated as per the suggestion

---

## [Decision Letter · Decision Letter 1]

1 Mar 2022

Knowledge, Attitude and Practice of Wearing Mask in the Population Presenting to Tertiary Hospitals in a Developing Country

PONE-D-21-33953R1

Dear Dr. furnaz,

We’re pleased to inform you that your manuscript has been judged scientifically suitable for publication and will be formally accepted for publication once it meets all outstanding technical requirements.

Kind regards,

Pathiyil Ravi Shankar

Academic Editor

PLOS ONE

Additional Editor Comments (optional):

Reviewers' comments:

Reviewer's Responses to Questions

**Comments to the Author**

1. If the authors have adequately addressed your comments raised in a previous round of review and you feel that this manuscript is now acceptable for publication, you may indicate that here to bypass the “Comments to the Author” section, enter your conflict of interest statement in the “Confidential to Editor” section, and submit your "Accept" recommendation.

Reviewer #1: All comments have been addressed

Reviewer #2: All comments have been addressed

2. Is the manuscript technically sound, and do the data support the conclusions?

Reviewer #1: Yes

Reviewer #2: Yes

3. Has the statistical analysis been performed appropriately and rigorously? 

Reviewer #1: Yes

Reviewer #2: Yes

4. Have the authors made all data underlying the findings in their manuscript fully available?

Reviewer #1: Yes

Reviewer #2: Yes

5. Is the manuscript presented in an intelligible fashion and written in standard English?

Reviewer #1: Yes

Reviewer #2: Yes

6. Review Comments to the Author

Reviewer #1: (No Response)

Reviewer #2: (No Response)

7. PLOS authors have the option to publish the peer review history of their article (what does this mean?). If published, this will include your full peer review and any attached files.

Reviewer #1: **Yes: **Rano Mal Piryani

Reviewer #2: **Yes: **Sikandar Ali Qalati

---

## [Editor Report · Acceptance letter]

3 Mar 2022

PONE-D-21-33953R1 

Knowledge, Attitude and Practice of Wearing Mask in the Population Presenting to Tertiary Hospitals in a Developing Country 

Dear Dr. Furnaz:

I'm pleased to inform you that your manuscript has been deemed suitable for publication in PLOS ONE. Congratulations! Your manuscript is now with our production department. 

Kind regards, 

on behalf of

Dr. Pathiyil Ravi Shankar 

Academic Editor

PLOS ONE